# Enhancement of Electromagnetic Wave Shielding Effectiveness by the Incorporation of Carbon Nanofibers–Carbon Microcoils Hybrid into Commercial Carbon Paste for Heating Films

**DOI:** 10.3390/molecules28020870

**Published:** 2023-01-15

**Authors:** Gi-Hwan Kang, Sung-Hoon Kim, Ji-Hun Kang, Junwoo Lim, Myeong Ho Yoo, Yi Tae Kim

**Affiliations:** 1Department of Energy and Chemical Engineering, Silla University, Busan 46958, Republic of Korea; 2Department of Research and Development, SH Korea Co. Ltd., Busan 49500, Republic of Korea

**Keywords:** controllable carbon microcoils, hybrid of carbon nanofibers-carbon microcoils, carbon paste, heating film, electromagnetic wave, shielding effectiveness, absorption shielding route

## Abstract

Carbon microcoils (CMCs) were formed on stainless steel substrates using C_2_H_2_ + SF_6_ gas flows in a thermal chemical vapor deposition (CVD) system. The manipulation of the SF_6_ gas flow rate and the SF_6_ gas flow injection time was carried out to obtain controllable CMC geometries. The change in CMC geometry, especially CMC diameter as a function of SF_6_ gas flow injection time, was remarkable. In addition, the incorporation of H_2_ gas into the C_2_H_2_ + SF_6_ gas flow system with cyclic SF_6_ gas flow caused the formation of the hybrid of carbon nanofibers–carbon microcoils (CNFs–CMCs). The hybrid of CNFs–CMCs was composed of numerous small-sized CNFs, which formed on the CMCs surfaces. The electromagnetic wave shielding effectiveness (SE) of the heating film, made by the hybrids of CNFs–CMCs incorporated carbon paste film, was investigated across operating frequencies in the 1.5–40 GHz range. It was compared to heating films made from commercial carbon paste or the controllable CMCs incorporated carbon paste. Although the electrical conductivity of the native commercial carbon paste was lowered by both the incorporation of the CMCs and the hybrids of CNFs–CMCs, the total SE values of the manufactured heating film increased following the incorporation of these materials. Considering the thickness of the heating film, the presently measured values rank highly among the previously reported total SE values. This dramatic improvement in the total SE values was mainly ascribed to the intrinsic characteristics of CMC and/or the hybrid of CNFs–CMCs contributing to the absorption shielding route of electromagnetic waves.

## 1. Introduction

The trend toward miniaturization and multifunctionality of electronic devices has exacerbated the problem of electromagnetic interference (EMI). As the applicable frequency range of electronic devices enters the higher frequency region, the shielding of electromagnetic wave radiation emitted from electronic devices is required to prevent not only the malfunction of other electronics, but also a significant threat to human health. This is because increased exposure can cause the device to malfunction and affect human health. The only solution to prevent damage from harmful radiation and protect electronics is to provide a shield that filters out the interference.

The shielding of electromagnetic waves typically proceeds via three main routes: reflection loss, absorption loss, and internal reflection loss [1]. For an absorption loss route greater than 10 dB, reflection and absorption loss routes are usually regarded as the main shielding routes [1,2,3,4]. In the relatively low-frequency range (typically less than 2.0 GHz), the reflection loss route is thought to be the crucial mechanism for preventing EMI [4,5]. However, at high operating frequencies (above 2.0 GHz), the absorption loss route is considered the main mechanism for preventing EMI [4,5]. High electrical conductivity is the key parameter for the shielding mechanism via the reflection loss route, while both high electrical conductivity and high magnetic permeability are important parameters for the shielding mechanism through the absorption loss route [1,5]. Therefore, in a relatively low frequency range, conductive metals, such as copper and aluminum, seem to be appropriate materials for EMI shielding via the reflection loss route. However, at higher operating frequencies, carbon-based materials would be the optimal materials through the absorption loss route because carbon-based materials can possess high magnetic field characteristics. Furthermore, carbon-based materials are lightweight and moldable [6,7,8]. Therefore, carbon-based materials are preferred as optimal materials for electromagnetic wave shielding in the relatively high operating frequency range for portable electronics, automobile electronics, and avionic electronic devices [5,9,10,11].

Recently, carbon-based heating films have been introduced as replacements for conventional heating systems [12,13,14,15,16,17]. They commonly consist of a substrate film made from polymers such as polyethylene terephthalate (PET). Next, they are coated with electrically resistive materials and are electrically contacted by busbars. The applicable materials for carbon-based heating films include carbon nanotubes (CNTs) [12,13], graphene oxide [14], reduced graphene oxide [15,16], and graphene nanoplatelets (GNPs) [17]. The materials used in busbars are commonly composed of silver, aluminum, and copper, and are laminated with another layer of insulating film, usually having similar characteristics to the substrate. Conductive polymers can occasionally be used together with carbon-based nanomaterials to improve the electrical conductivity of carbon-based films [18]. Thus, carbon-based heating films are made of stacks comprised of many individual functional films, which can give rise to efficient heat-dissipating capabilities. This heating film system has several advantages, such as energy savings, eco-friendliness, and durability. A difficult problem in the application of a heating film is the shielding of electromagnetic waves emitted from the heating source in the film.

To date, there are some reports of heating films being used to shield against electromagnetic interference. Lee et al. registered a patent concerning a film-type planar heating element that can be used to prevent electromagnetic waves from generating heat in the form of multiple layers of films [19]. Kim et al., registered a patent for a production method of heating films to be used in electromagnetic wave shielding applications [20]. Kang et al.suggested a method for generating heat on a large-area graphene film more efficiently by utilizing the unique electromagnetic wave absorption properties of graphene [21].

Lin et al. showed that graphene exhibits a higher electron shielding effect than CNT. In addition, it was shown that when CNT and graphene were hybridized, electrical conductivity was improved by about eight times [22]. In previous research results, materials with a three-dimensional structure, compared to one-dimensional shielding materials, have higher electromagnetic wave shielding effects [23]. We introduce carbon microcoils (CMCs)-related materials, as a three-dimensional structure, in the manufacturing of heating films which can be used as shielding agents against electromagnetic waves. CMCs exhibit unique helical geometries [24]. When an incoming electromagnetic wave reaches the CMCs, electric current flows through helically oriented individual carbon fibers (CFs) situated on the CMCs, thereby inducing an electromotive force and generating a variable magnetic field [25,26,27]. Finally, the incoming electromagnetic wave energy is absorbed into the unique geometries of CMCs and converted into thermal energy [26]. Previously, we reported an increase in the shielding effectiveness (SE) of the CMCs–polyurethane composites with increasing content of CMCs [28]. Kim et al. reported that the SE of a hybridized carbon microcoil–carbon fiber nonwoven fabric increased slightly [29]. In addition, diverse hybrid formations using carbon-based materials have been developed to enhance the SE values [29,30,31]. These combined results clearly suggest that the formation of CMCs-related materials could enhance SE values.

In the present work, controllable CMCs were obtained by manipulating the additive gas, SF_6_, and flow injection time. In addition, the hybrid formation of numerous small carbon nanofibers (CNFs) on the surfaces of CMCs (CNFs–CMCs) can be achieved by the injection of H_2_ flow with the cyclic process of SF_6_ flow. The SE of the heating film made from the CNFs–CMCs hybrid material incorporated with carbon paste was investigated across the operating frequencies in the 1.5–40 GHz range. The results were compared with those for heating films made using a native commercial carbon paste or controllable CMCs-incorporated carbon paste. The morphologies and electrical conductivities of different types of heating films were investigated. The main shielding mechanism of the heating film made from the hybrids of CNFs–CMCs incorporated carbon paste is suggested and discussed.

## 2. Results and Discussion

Figure 1 shows the magnified FESEM images of the surface morphologies for samples A–I. Under a flow rate of 20 sccm SF_6_, the diameters of the CMCs increased with the SF_6_ flow injection time (see Figure 1a–c). Under the 50 sccm and 100 sccm SF_6_ flow rates, the CMCs diameters also tended to increase with increasing SF_6_ flow injection time (see Figure 1d–f and Figure 1g–i, respectively). This reveals that the lowest SF_6_ flow injection time in this study (5 min) can give rise to the CMCs with the smallest diameters, irrespective of the SF_6_ flow rate. In addition, Figure 1 shows that overall the diameters of the CMCs are independent of the SF_6_ flow rate. These results strongly suggest that the SF_6_ flow injection time, instead of the SF_6_ flow rate, can directly influence the formation of CMCs with a specific diameter.

Figure 2 shows FESEM images of sample J, which were obtained by the injection of H_2_ flow and the cyclic process of SF_6_ flow. As shown in Figure 2a, a woolen yarn-type surface is present in sample J. The magnified FESEM image of sample J also indicated the existence of many small-sized CNFs on the CMCs surfaces (see Figure 2b,c). As previously reported, the small-sized CNFs around the CMCs seem to have a hybridized aspect between the CNFs and CMCs by the cyclic process of SF_6_ flow [32,33,34,35,36,37,38,39,40]. This result suggests that the cyclic SF_6_ flow with the injection of H_2_ gas could produce CNFs–CMCs hybrids [40].

The presence of different nanocarbon formation attributes in sample J compared to samples A–I can be explained as follows. It is known that producing a hybrid nanocarbon from typical nanocarbon materials, such as CNTs with CNFs and CMCs with carbon nanocoils, is very difficult because the transition metals used as catalysts to promote hybrid nanocarbon growth tend to easily diffuse into the interior of the carbon substrate during the reaction [41,42]. For the surfaces of CMCs during the reaction, the tiny Ni catalysts used for hybrid nanocarbon growth would not be sufficient because tiny Ni catalysts could easily diffuse into the interior of CMCs in an amorphous solid state [43,44,45]. Consequently, numerous small-sized CNFs could not be formed on the surfaces of the CMCs of samples A–I because of the lack of tiny Ni catalysts on the surfaces of the CMCs during the reaction.

However, a previous report revealed that the injection of H_2_ gas into the flow of C_2_H_2_, and an abundant C_2_H_2_ gas flow relative to the SF_6_ gas flow in the reaction environment are known to facilitate the formation of numerous small-sized CNFs by the generation of numerous tiny Ni catalysts from large-sized Ni catalysts [46]. In this work, the H_2_ flow injection for sample J was subjected to the injection of H_2_ flow into the C_2_H_2_ flow. Furthermore, the cyclic process in sample J could produce an abundant C_2_H_2_ flow environment compared with SF_6_ gas flow, especially during the SF_6_ flow-off period [40]. Compared to samples A–I, sample J produced a much higher number of tiny Ni catalysts, which could readily be placed on the surfaces of the CMCs. Consequently, this would result in the formation of small-sized CNFs on the surfaces of the CMCs owing to the increased number of tiny Ni catalysts, probably enough to overcome the insufficiency of Ni catalysts on the surfaces of the CMCs. This can be attributed to the diffusion of Ni catalysts into the interior of the CMCs during the reaction.

Figure 3 shows systematic diagrams for the formation of samples A–I (C_2_H_2_ + SF_6_ gas flow system without cyclic process of SF_6_ gas flow, Figure 3a) and sample J (C_2_H_2_ + SF_6_ + H_2_ gas flow system with a cyclic process of SF_6_ gas flow, Figure 3b). Ni fragments were located merely on the heads of CMCs for samples A–I (Figure 3a). However, for Sample J, Ni fragments were present on the CMC head and the tiny-sized Ni fragments were present on the CMC surface. Moreover, the numerous tiny Ni fragments, produced by the injection of H_2_ gas into the C_2_H_2_ flow during the cyclic addition of SF_6_, were present on the surfaces of the CMCs as shown in Figure 3b.

For the mass production of CMCs-related nanocarbon samples, we chose the conditions of samples E and J among the various types of CMCs-related nanocarbon formation reaction conditions. Almost 19 g of CMCs per approximately 0.6 g Ni catalyst could be obtained in a one-batch reaction, as shown in Figure 4a. Figure 4b–d show the FESEM images of these CMCs samples. The obtained sample seemed to have CMCs with fixed diameters within the range of 1–5 μm. The magnified FESEM image of this sample (Figure 4d) clearly indicated the formation of well-developed controllable CMCs geometries under these reaction conditions.

Under the experimental conditions of sample J, approximately 20 g of the hybrids of CNFs–CMCs per approximately 0.6 g Ni catalyst could be obtained in a one-batch reaction, as shown in Figure 5a. Figure 5b–d clearly show the formation of hybridized CNFs–CMCs between the numerous small-sized CNFs and CMCs.

After achieving mass production of CMCs-related materials, we made a blend of 30 wt% sample J with 70 wt% commercial carbon paste. A blend of 30 wt% sample E with 70 wt% commercial carbon paste was also prepared to compare the SE values of samples J and E. These blends were coated on a PET film by a commonly used printing method using a spatula blade. Figure 6 shows the SE values of these coated PET films across the operating frequencies in the 1.5–40 GHz range.

Compared with the total SE values of the PET film coated with 30 wt% sample E-incorporated carbon paste (see Table 1), the value of the PET film coated with 30 wt% sample J-incorporated carbon paste was more than two-fold higher in dB scale across the entire operating frequency range, as shown in Figure 6.

This dramatic increase in the total SE values of the coated PET film by 30 wt% sample J-incorporated carbon paste seems to be partly ascribed to the enhanced electrical conductivity (from (1.84 ± 0.46) × 10^3^ S/m to (2.17 ± 0.23) × 10^3^ S/m, see Table 2) by the hybrid formation between the numerous small-sized CNFs and the CMCs in sample J, as in a previous report [30]. Furthermore, the numerous small-sized CNFs on the surfaces of the CMCs in sample J intersected with one another. When an incoming electromagnetic wave reaches these intersected-CNFs, electric current flows into the intersections and dissipates in various directions, thereby inducing an electromotive force and generating a variable magnetic field [25]. The geometry of these CNFs holds and rotates incoming electromagnetic waves within the generated variable magnetic field. Thus, the incoming electromagnetic wave energy is absorbed into these CNFs and is finally converted into thermal energy [26]. Therefore, these intersections can contribute to the absorption mechanism for shielding against electromagnetic waves. Consequently, the total SE values of the PET film coated with the 30 wt% sample J-incorporated carbon paste were higher than those of the 30 wt% sample E-incorporated carbon paste. This matches the measured SE values shown in Figure 6. The skin depth (δ) of a shielding material is defined as δ = (πσ*f*µ)^−1/2^ [2], indicating that δ^2^ is inversely proportional to the electrical conductivity (σ), frequency (*f*), and magnetic permeability (µ). Therefore, higher magnetic permeability can efficiently reduce the skin depth of the shielding material, thereby enhancing the SE values. The intrinsic characteristics of CMCs, and the aspect of numerous small-sized CNFs intersecting with one another in sample J, can enhance the magnetic field and then absorb incoming EM waves. Consequently, they can enhance magnetic permeability (µ), resulting in an improvement in the absorption loss of electromagnetic waves. Therefore, the SE values of the PET film coated with the 30 wt% sample J-incorporated carbon paste were much higher than those of the 30 wt% sample E-incorporated carbon paste across the entire operating frequency range.

Indeed, the PET film coated with 30 wt% sample J-incorporated carbon paste had total SE values above 20 dB throughout the entire range of operating frequencies. Compared with the previously reported total SE values, the presently measured values seem to be ranked in the top tier (see Table 2). Therefore, we suggest that the hybridized CNFs–CMCs sample can be effectively used in diverse industrial fields.

A blend of 2 wt% sample J with 98 wt% commercial carbon paste was also prepared to determine the optimal amount of sample J to add to the commercial carbon paste. Indeed, the already established manufacturing process for the commercial heating film by SH Korea Co. required the injection of the smallest possible amount of sample J, because a considerable amount of sample J may cause poor adhesion between the paste layer and the PET substrate. The PET film coated with the 2 wt% sample J-incorporated carbon paste seemed to satisfy the adhesion problem, as well as provide good SE values, as shown in Figure 6.

For the manufacturing process of the commercial heating film, we used a coated PET film with 5 wt% sample J-incorporated carbon paste. In this case, we used a typical laminating process using the gravure method of SH Korea Co. The typical thickness of the coated layer using carbon paste was approximately 50 μm. Basically, a single des-HF was composed of two electromagnetic wave shielding layers located at the front and back sides among eight different functional individual thin layers. Therefore, we estimated a total thickness of 100 μm for the coated layers using 5 wt% sample J-incorporated carbon paste.

Figure 7 shows the total SE values for the conventional heating film using native commercial carbon paste, des-HF manufactured by 5 wt% sample E-incorporated carbon paste, and des-HF manufactured by 5 wt% sample J-incorporated carbon paste. As discussed in the results depicted in Figure 6, the total SE values of the native conventional heating film using commercial carbon paste were enhanced by the incorporation of 5 wt% sample E, and further enhanced by the incorporation of 5 wt% sample J. The cause for the improvement of the SE values seems to be largely attributable to the fact that numerous small CNFs intersected with one another and/or the intrinsic characteristics of CMCs; however, the electrical conductivity of the coated PET film using the commercial carbon paste was decreased by the incorporation of controllable CMCs, or hybrids of CNFs–CMCs in the commercial carbon paste (see Table 1).

Figure 8 shows the total SE values, the SE values for the absorption loss, and the SE values for the reflection loss of the des-HF manufactured by 5 wt% sample J-incorporated carbon paste across the operating frequencies in the 1.5–40 GHz range. Above the 4.0 GHz frequency range, the absorption SE values of the des-HF manufactured by 5 wt% sample J-incorporated carbon paste increased and approached the values of the total SE values, as shown in Figure 8. This confirms that the higher total SE values of the hybrids of CNFs–CMCs in this work are mainly attributable to the enhanced absorption loss across the operating frequencies in the 4.0–40 GHz range. Therefore, the enhanced total SE values obtained using the hybrids of CNFs–CMCs were mainly ascribed to the enhanced absorption shielding loss, contributed by the intrinsic characteristics of CMCs and the aspect of numerous small-sized CNFs intersecting with one another in sample J.

## 3. Materials and Methods

As a catalyst for the formation of CMCs, approximately 0.1 g of bunch-type Ni powder (99.7%), with particle diameters ranging from 1 μm to 10 μm, was spread onto a 2 mm-thick, boat-like stainless steel (SUS304) substrate. A thermal chemical vapor deposition (CVD) system was employed for the formation of CMCs, using C_2_H_2_ as the source gas and SF_6_ as the additive gas. The deposition reaction conditions for the formation of the various CMCs-related samples are listed in Table 3. Ten samples (samples A–J) with different combinations of gas flow rate, gas flow injection time, and gas type were prepared.

Regarding the application of SF_6_ gas in sample J, the cyclic process was conducted by simply switching the SF_6_ flow on and off continuously. The gas flow sequence mirrored the iterative order of the reaction processes: C_2_H_2_ + H_2_ + SF_6_ flow (C_2_H_2_ flow-on, H_2_ flow-on, and SF_6_ flow-on) followed by C_2_H_2_ + H_2_ flow (C_2_H_2_ flow-on, H_2_ flow-on, and SF_6_ flow-off), as shown in Figure 9. The cycle period was defined as the sum of the time the source gases were composed of C_2_H_2_ + H_2_ + SF_6_ flow, and the time the source gases consisted solely of C_2_H_2_ + H_2_ flow. For sample J, the on and off times for the SF_6_ flow injection were set at 1.5 min, resulting in a total duration of 3.0 min for one cycle. Because the total cyclic on/off modulation of the SF_6_ flow was 15 min, five cycles were performed during the reaction.

The morphologies of the CMCs samples were investigated in detail by field emission scanning electron microscopy (FESEM; S-4200 Hitachi, Tokyo, Japan). The thickness of the sample was measured using a micrometer (406-250-30 Mitutoyo, Nakagawa, Japan) and corrected using cross-sectional FESEM images. Resistivity values were obtained by using a four-point probe (labsysstc-400 Nextron, Busan, Republic of Korea) connected to a source meter (2400 Source Meter Keithley, Cleveland, OH, USA) and by performing calculations using Ohm’s law with a correction factor, according to the method proposed by Smits [32]. The four-point probe system consisted of four, straight-lined probes with an equal inter-probe spacing of 3.0 mm. A constant current (*I)* was supplied through the two outer probes, and the output voltage (*V*) was measured using the two inner probes [31]. Correction factors (*C* and *F*) were obtained from Smits et al. [32]. Surface and volume resistivities were calculated using the following equations [32,33]:$$\mathrm{Surface}\;\mathrm{resistivity}:\;\rho_{s}=\frac{V}{I}C\left(\frac{a}{d},\frac{d}{s}\right){,}_{\;}\mathrm{volume}\;\mathrm{resistivity}:\;\rho_{v}=\rho_{s}\;w\;F\;\left(\frac{w}{s}\right)$$
where *a*, *d*, *w*, and *s* denote the length, width, and thickness of the sample and the inter-probe spacing, respectively.

The final product of the heating films, namely dual electromagnetic shielding premium heating films (des-HFs), were prepared by a typical laminating process with the gravure method of SH Korea Co. (see Figure 10) using the blends of CMCs-related samples and commercially supplied carbon paste. 2-butoxyethyl acetate was used as the diluting solution during the gravure coating process with these blends.

A single des-HF film was composed of eight different functional individual thin films, as shown in Figure 11.

The SE values of the des-HFs were measured using a waveguide method with a vector network analyzer (VNA; 37369C Anritsu, Kanagawa, Japan), as shown in Figure 12. The results were compared with those of a heating film made using commercially available carbon paste (CKC-300 AF Electrochem, Incheon, Republic of Korea). The setup for the VNA system consisted of a sample holder with its exterior connected to the system. A coaxial sample holder and a coaxial transmission test specimen were set up according to the waveguide method. The scattering parameters (S_11_ and S_21_) were measured in the frequency range of 1.5−40 GHz using the VNA [34,35,36,37,38]. The power coefficients, namely reflectivity (*R*), absorptivity (*A*), and transmissivity (*T*), were calculated using the following equations: *R* = *P*_R_/*P*_I_ = |S_11_|^2^ and *T* = *P*_T_/*P*_I_ = |S_21_|^2^, where *P*_I_, *P*_R_, *P*_A_, and *P*_T_ are the incident, reflected, absorbed, and transmitted powers of an electromagnetic wave, respectively [38]. The power coefficient relationships were expressed as *R* + *A* + *T* = 1. The SE of the electromagnetic waves was calculated from the scattering parameters using the following equation:*SE*_Tot_ = −10 log *T*,(1)
*SE*_R_ = −10 log (1 − *R*),(2)
*SE*_A_ = −10 log{*T*/(1 − *R*)},(3)
where, *SE*_Tot_, *SE*_R_, and *SE*_A_ denote the total, reflection, and absorption SE values, respectively [37,38].

## 4. Conclusions

Controllable CMCs, especially the controlled-diameter size of CMCs, could be achieved by manipulating the injection gas parameters under C_2_H_2_ + SF_6_ gas flow in a thermal chemical vapor deposition system. The diameters of the CMCs were independent of the SF_6_ flow rate, but strongly dependent on the SF_6_ flow injection time. The cyclic process of SF_6_ flow with the injection of H_2_ flow could produce the formation of CNFs–CMCs hybrid. The formation of the hybrids of CNFs–CMCs by the cyclic process of SF_6_ flow with the injection of H_2_ flow was explained by the generation of a large number of tiny Ni catalysts to overcome the deficiency of tiny Ni catalysts on the surfaces of the CMCs. This was attributed to the facile diffusion of Ni catalysts into the interior of the CMCs during the reaction. Systematic diagrams were deployed to assist the different formation of the controllable CMCs and hybrids of CNFs–CMCs, according to the different gas injection aspects.

For one batch reaction, approximately 20 g of the hybrids of CNFs–CMCs with numerous small-sized CNFs around the CMCs was obtained by about 0.6 g Ni catalyst onto a 2mmthick, boat-like stainless steel (SUS304) substrate.

Compared with the total SE values of the PET film coated with 30 wt% controllable CMCs-incorporated carbon paste, the values of the PET film coated with 30 wt% hybrids of CNFs–CMCs incorporated carbon paste were more than two-fold higher in dB scale across the entire operating frequency range. This dramatic increase in the total SE values was partly ascribed to the enhanced electrical conductivity, due to the hybrid formation between the numerous small CNFs and the CMCs. Furthermore, the intrinsic characteristics of CMCs and the behavior of numerous small-sized CNFs intersecting with one another in the CNFs–CMCs hybrids can enhance the magnetic field and absorb incoming EM waves. Consequently, they can improve the absorption loss of electromagnetic waves. Therefore, the SE values of the PET film coated with the hybrids of CNF-CMCs incorporated carbon paste were much higher than those with 30 wt% controllable CMCs incorporated carbon paste across the entire operating frequency range.

Although the electrical conductivity of the coated PET film using the commercial carbon paste was decreased by the incorporation of the hybrids of CNFs–CMCs in the commercial carbon paste, the total SE values of the conventional heating film using the commercial carbon paste were significantly enhanced by the incorporation of 5 wt% hybrids of CNFs–CMCs. The cause for this dramatic enhancement of the SE could be attributable to the fact that numerous small-sized CNFs intersect with one another and the intrinsic characteristics of CMCs.

## Figures and Tables

**Figure 1 molecules-28-00870-f001:**
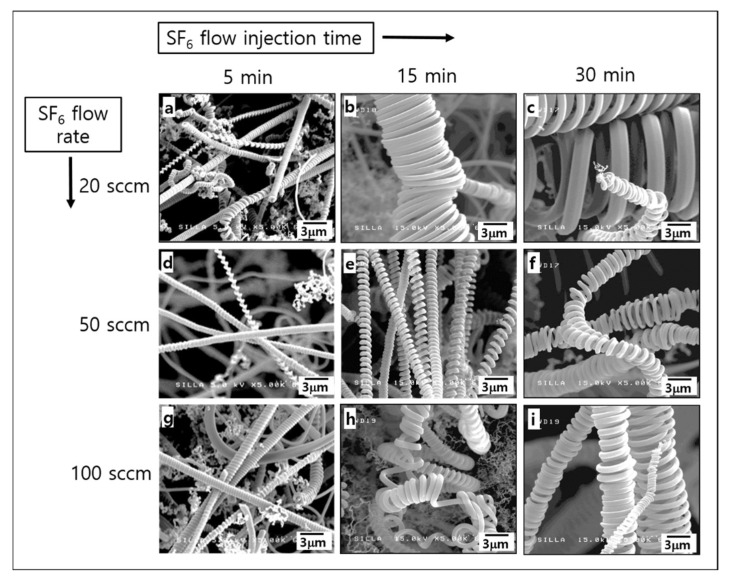
The FESEM images of the surface morphologies for samples (**a**) A, (**b**) B, (**c**) C, (**d**) D, (**e**) E, (**f**) F, (**g**) G, (**h**) H, and (**i**) I.

**Figure 2 molecules-28-00870-f002:**
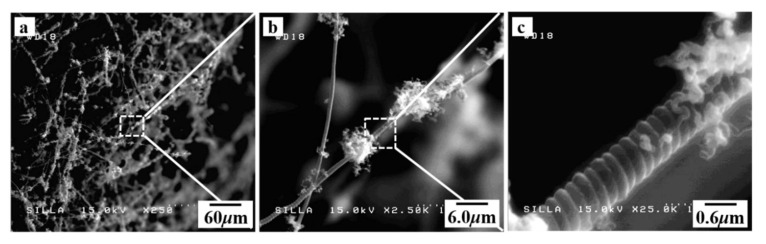
(**a**) The FESEM image of the surface morphology for sample J, (**b**) the magnified-FESEM image of the squared-area in (**a**), and (**c**) the high magnified-FESEM image of the squared-area in (**b**).

**Figure 3 molecules-28-00870-f003:**
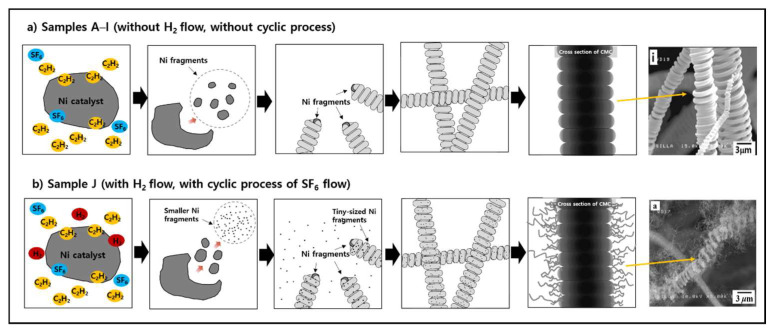
Systematic diagrams indicating (**a**) the formation of controllable CMCs in the C_2_H_2_ + SF_6_ gas flow system and (**b**) the formation of numerous small-sized CNFs on the surfaces of CMCs caused by H_2_ flow injection and the abundant C_2_H_2_ flow during SF_6_ flow off period in the cyclic process.

**Figure 4 molecules-28-00870-f004:**
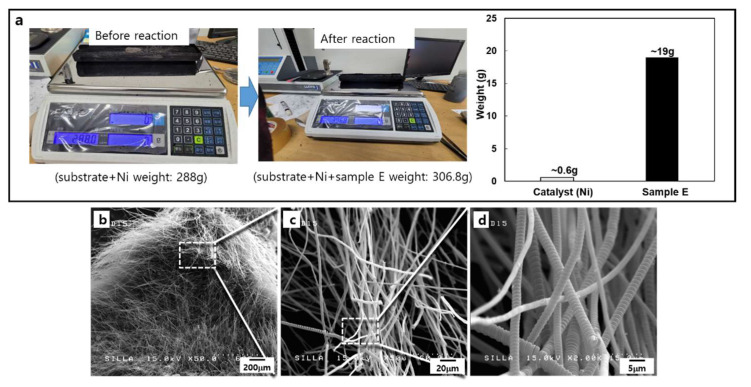
(**a**) Optical photograph for the weighting of sample E, (**b**) FESEM image of sample E, (**c**) magnified-FESEM image of (**b**), and (**d**) high magnified-FESEM image of (**c**).

**Figure 5 molecules-28-00870-f005:**
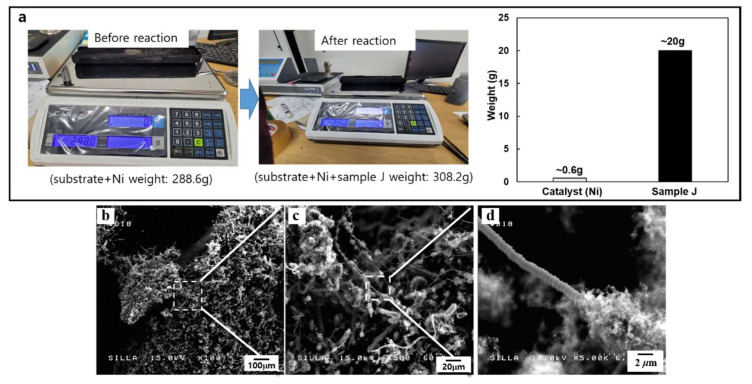
(**a**) Optical photograph for the weighting of sample J, (**b**) FESEM image of sample E, (**c**) magnified-FESEM image of (**b**), and (**d**) high magnified-FESEM image of (**c**).

**Figure 6 molecules-28-00870-f006:**
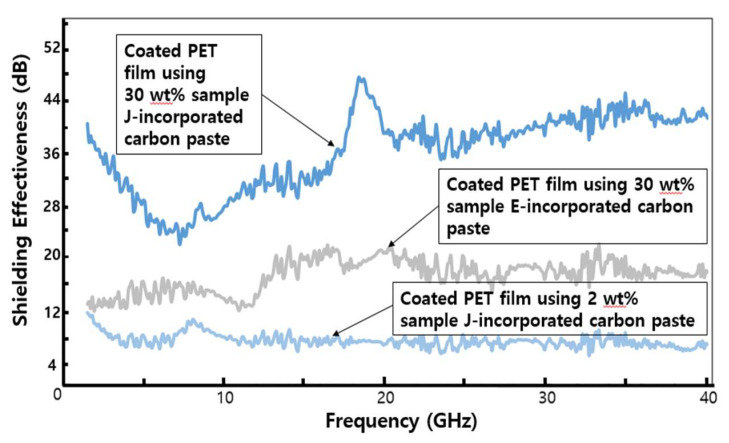
The total SE values of the coated PET films using the blends of CMCs-related materials and commercial carbon pastes.

**Figure 7 molecules-28-00870-f007:**
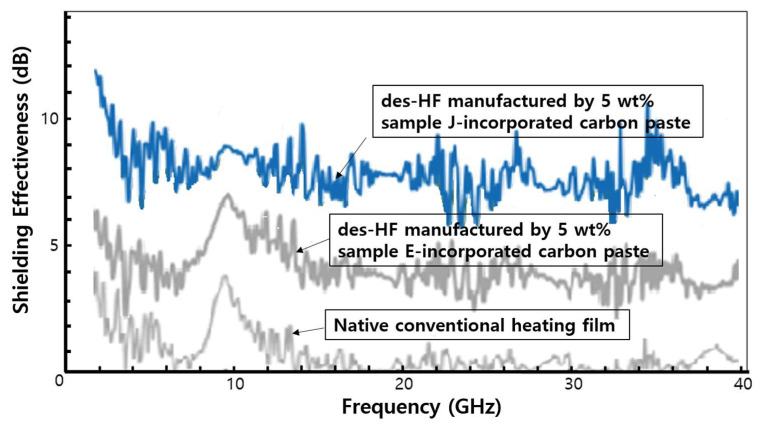
The total SE values for the conventional heating film using native commercial carbon paste, the des-HF manufactured by 5 wt% sample-E incorporated carbon paste, and the des-HF manufactured by 5 wt% sample-J incorporated carbon paste.

**Figure 8 molecules-28-00870-f008:**
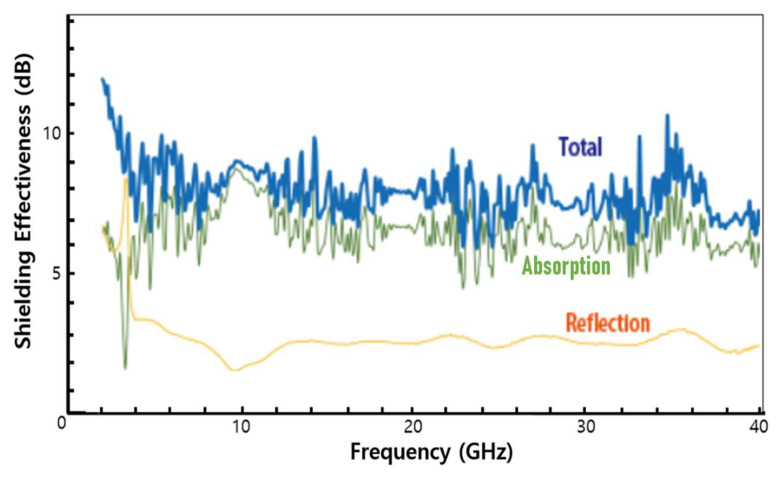
Total SE values, the SE values for the reflection loss, and the SE values for the absorption loss of the des-HF manufactured by 5 wt% sample J-incorporated carbon paste.

**Figure 9 molecules-28-00870-f009:**
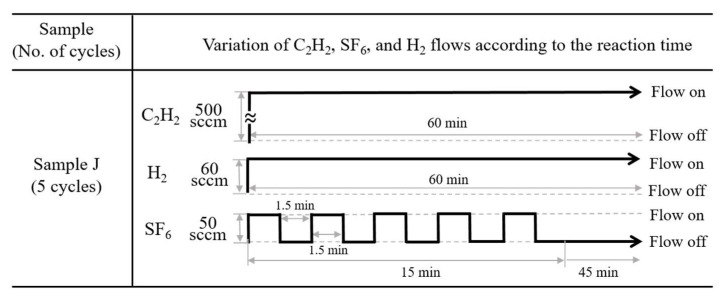
Cyclic injection processes of C_2_H_2_, H_2_, and SF_6_ flows for sample J.

**Figure 10 molecules-28-00870-f010:**
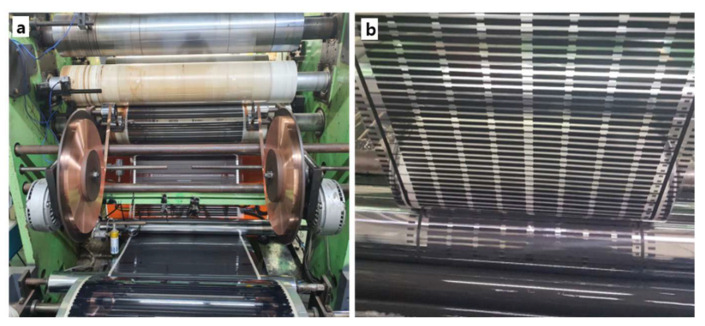
The production of heating film: (**a**) the coating equipment used to perform the gravure method and (**b**) the actual gravure coating process of carbon paste on the base film.

**Figure 11 molecules-28-00870-f011:**
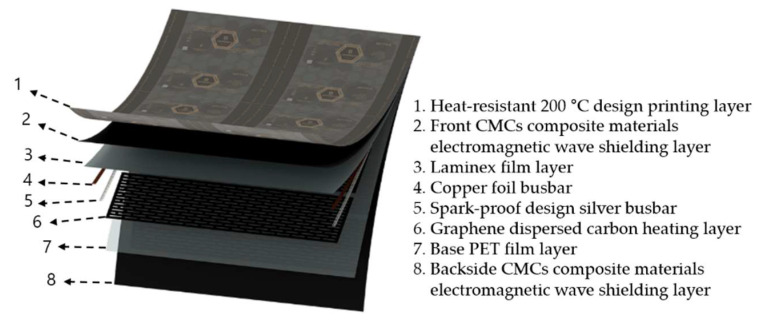
Structure of dual electromagnetic shielding premium heating film.

**Figure 12 molecules-28-00870-f012:**
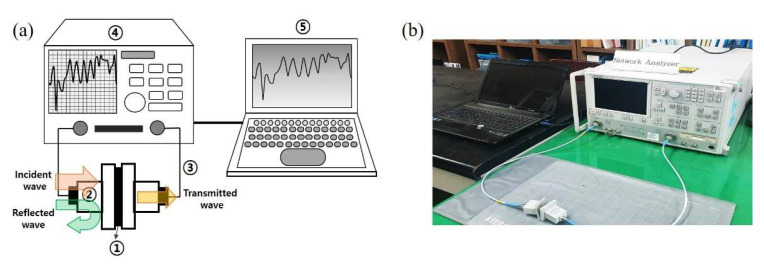
(**a**) Schematic of the vector network analyzer (VNA): ① sample, ② waveguide test holders, ③ coaxial cables, ④ VNA, and ⑤ computer and (**b**) optical photograph of the experimental setup for the EMI shielding measurements.

**Table 1 molecules-28-00870-t001:** Thickness, electrical resistivity, and electrical conductivity of the coated PET films by commercial carbon paste, 30 wt% sample E-incorporated carbon paste, and 30 wt% sample J-incorporated carbon paste.

The Coated PET Films and Type of Carbon Paste	Thickness *t* (mm)	Electrical Resistivity *ρ*(Ω·m)	Electrical Conductivity *σ* (S/m)	Correction Factor * *F* (F/w)
Commercial carbon paste	0.6 (±0.04)	2.42 (±0.15) × 10^−4^	4.13 (±0.08) × 10^3^	0.98
30 wt% sample E-incorporated carbon paste	0.6 (±0.07)	5.43 (±1.22) × 10^−4^	1.84 (±0.46) × 10^3^	0.79
30 wt% sample J-incorporated carbon paste	0.6 (±0.05)	4.61 (±0.65) × 10^−4^	2.17 (±0.23) × 10^3^	0.79

* Correction factor was calculated from Table 2 in Ref. [32].

**Table 2 molecules-28-00870-t002:** EMI SE of carbon-based materials.

Carbon-Based Materials	Thickness (mm)	Electrical Conductivity or Resistivity	Operating Frequency (GHz)	SE (dB)	SE/Thickness (dB/mm)	Ref.
CNT/*MCMB	0.15–0.6	1100 S/m	8.2–12.4	31–56	93–206	[47]
15 wt% *CNF/*ABS	1.1	1.5 ± 0.1 Ω·cm		35	31.8	[48]
15 wt% CNT/*ABS		0.81 ± 0.05 Ω·cm		51	46.4	
*CNF/epoxy	2.1	-		5–34	2.4–16.2	[49]
GNP/*PEDOT:PSS	0.8	684 S/m		70	88	[50]
*SCF/*EVA	3.5	-	8–12	29.5–34.1	8.4–9.7	[51]
*MX/*RGO	3	1000 S/m		51	17	[52]
*3D G–CNT–Fe_2_O_3_	0.6	22,781 S/m		130–134	216–223	[53]
*GN/Cu	0.009 (±0.0015)	5.88 (±0.29) × 10^6^ S/m	1–18	52–63	5777–7000	[54]
*SWCNT/epoxy	1.5	20 S/m	0.01-1.5	15–49	10–32.6	[55]
The coated PET film using 30 wt% sample E-incorporated carbon paste	0.6	1840 S/m	1.5~40	12–24	20–40	This work
The coated PET film using 30 wt% sample J-incorporated carbon paste	0.6	2170 S/m		24–56	40.0–93.3	

*CNT: carbon nanotube, *MCMB: mesocarbon microbeads, *CB: carbon black, *CNF: carbon nanofiber,*ABS: acrylonitrile–butadiene–styrene, *PEDOT:PSS: poly(3,4-ethylenedioxythiophene)–poly(styrene-sulfonate), *SCF: short carbon fiber, *EVA: ethylene vinyl acetate, *MX: Mxene, *RGO: reduced graphene oxide, *3D G–CNT–Fe_2_O_3_: three-dimensional graphene–carbon nanotube–iron oxide, *GN: graphene, *SWCNT: single wall carbon nanotube.

**Table 3 molecules-28-00870-t003:** Experimental conditions for Samples A–J.

Sample	C_2_H_2_Flow Rate (sccm)	H_2_ Flow Rate (sccm)	SF_6_ Flow Rate (sccm)	SF_6_Flow Injection Time (min)	No. of SF_6_ Flow On/Off cycles	Total Reaction Time (min)	Total Gas Pressure (Torr)	Substrate Temp. (°C)	Remarks
A	500	-	20	5	-	60	100	750	Without cyclic process
B	500	-	20	15	-	60	100	750	Without cyclic process
C	500	-	20	30	-	60	100	750	Without cyclic process
D	500	-	50	5	-	60	100	750	Without cyclic process
E	500	-	50	15	-	60	100	750	Without cyclic process
F	500	-	50	30	-	60	100	750	Without cyclic process
G	500	-	100	5	-	60	100	750	Without cyclic process
H	500	-	100	15	-	60	100	750	Without cyclic process
I	500	-	100	30	-	60	100	750	Without cyclic process
J	500	60	50	15	5	60	100	750	With cyclic process of SF_6_ flow

## Data Availability

All data used and/or analyzed during the current study are shown within the study. If further data are required, theycan be made available by the corresponding author upon reasonable request.

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
