# Peer review of "Enhancement of Electromagnetic Wave Shielding Effectiveness by the Incorporation of Carbon Nanofibers–Carbon Microcoils Hybrid into Commercial Carbon Paste for Heating Films"

_molecules, 2023, doi:10.3390/molecules28020870_

Round 1

Reviewer 1 Report

Authors reported the shielding effectiveness of various carbon microcoils (CMCs) fabricated by controlling the flow rate and the injection time of SF6 gas with CVD technique in this article. And they showed that the hybrid carbon CNFs-CMCs has higher SE value than carbon micorcoils due to the CNF formed around the carbon micorcoils. In addition, they tested the SE value of PET film coated with the hybrid CNFs-CMC for the commercial application.

1)      I wonder what is the novelty of this article.  Authors have published the similar results in Applied Surface Science 2019.

2)      In the Caption of the Figure 1, remove (a)

3)      Line 201 : Authors mentioned the existence of many small-sized CNFs around the CMCs surfaces for the sample J. However, a lot of CNFs-like are observed around the CMC as shown in Fig. 5,  b, J, h, as well. What is the sizes of the CNFs-like observed in b, J, h and the CNFs around CMCs in the sample J ? For the comparison of their size, Authors need to notate the sizes of the CMCs for sample A through J.

4)      Authors claim that the SF6 flow rate does not influence the diameter of the CMCs during the reaction. I see a lot of small sized CMC bundle (or CNFs) around the micro sized CMCs (SEM images in Fig. 5). The SF6 flow rate may affect the uniformity in the size of the CMC. Think about this.  

5)      In discussion, Authors claim that the tiny Ni catalysts easily diffuse into the interior of CMCs in an amorphous solid state for the samples A –I, but for the sample J, the small sized Ni fragments do not diffuse into the interior of CMCs. Can you explain why the tiny Ni fragments do not diffuse into the CMCs for the sample J ? is it the lack of SF6 due to the off-time of SF6 or the effect of H2 gas during the reaction ?

6)      Line 238 - 242. Need to check the sentences. It might be something like that “for sample J, Ni fragments were present on the heads of CMC with the numerous tiny sized Ni fragments - - - ”

7)      Line 283 – 285, Authors claim that the dramatic increase in the total SE values of the coated PET film by 30 wt% sample J-incorporated carbon paste seems to be partly ascribed to the enhanced electrical conductivity due to the hybrid formation between the numerous small CNFs and the CMCs. The conductivity of the PET film coated with commercial carbon paste is the highest as 4.13 x 103 in Table 2. What is the SE value of the PET film coated with carbon paste ? Authors need to add the SE value as a control to the Fig. 10.  

Reviewer 2 Report

Kang, Kim et al. report on the CVD-growth of Carbon microcoils and their potential use for electromagnetic shielding. The paper is well structured, the synthesis is well documented by analytic results merged into graphical items such as carefully annotated viewgraphs and mindmaps with sufficiently large font sizes and good readibility. The results are well rooted in the literature, shown by dilligent and rigorous analysis and tabular summaries. As a minor monitum, the introduction of the electromagnetic basics (such as EMI, SE) as an anchor for the authors' storyline and the relations to other low-dimensional carbon nanomaterials concepts such as fibres, bundles, yarns etc. to the interdisciplinary audience as well as a first-order modeling idea (SE(f) = P(f) + n0) could deserve minor upgrades.

lines 36/37: The sentence "For an absorption loss route greater than 10 dB, reflection and absorption loss routes are usually regarded as the main shielding routes [14]." seems to include a circular reference regarding "absorption loss" and the quantity 10 dB seems to be chosen without sufficient reasoning: Please specify "absorption". Please consider the figure 10 dB as a function of distance, solid angle etc.

lines 40 ff: The role of electromagnetic interference (EMI) requires a more pronounced introduction for the interdisciplinary readership. Please write why this is important to look at.

lines 54 ff: I would expect a simple viewgraph here that correlates operating frequency ranges and materials systems including their degree of usage.

line 63 ff: The section on carbon nanomaterials could be enhanced by digging into the libraries of low-dimensional (1D, 2D) carbon nanotube solid materials concepts and relating their potential functional properties: bundles, fibres, yarns, networks and thin films, cf. e.g. Wagner et al. PSSA 2019 DOI 10.1002/pssa.201900584. The role of CMCs and CNF-CMCs can then be discriminated.

line 84: At this point, a more detailed introduction of the "shielding effectiveness" (SE) as a quantity appears to be crucial for the understanding of the experimental approach.

line 169 f: The equation appears as three equations, unless changed to vector form. Please correct and/or use Eq. number(s).

line 273/337/353: Figure 10/11/12: The experimentalist's view asks why (not) to apply a suitable curve fitting to the graphs, e.g. binning, nearest neigbour or Savitzky-Golay fitting and/or mathematical modelling to the form SE(f) = n0 + P(f) where P is a polynomial (or other) function. In the later text, I find the oscillations in the sub-GHz regime not discussed, rather is reference given to the offset values (n0_J1, E, J2).

line 353: Fig. 12: Please correct label "Absorption"

line 421: Please consider line feed or section change to correct spacing.

#
